# The Molecular Pathology of Pre-Eclamptic Hypertension

**DOI:** 10.3390/cimb47050375

**Published:** 2025-05-20

**Authors:** Robin W. Carrell, Randy J. Read, Aiwu Zhou

**Affiliations:** 1Department of Haematology, Cambridge Institute for Medical Research, Trinity College, University of Cambridge, Cambridge CB2 1TQ, UKawz20@shsmu.edu.cn (A.Z.); 2Institute for Translational Medicine on Cell Fate and Disease, Department of Pathophysiology, Shanghai Jiao Tong University School of Medicine, Shanghai 200025, China

**Keywords:** angiotensinogen, hypertension, pre-eclampsia, S-nitrosylation, placenta redox stress

## Abstract

The central role of angiotensinogen in the control of blood pressure is revealed by a series of crystallographic structures, including complexes with renin. Specifically, the structures provide an understanding of the sequential molecular events that lead to the pre-eclamptic hypertensive crises of pregnancy. The release of the precursor vasopressor peptide from the amino-terminal tail of angiotensinogen appears to be modulated by a redox-sensitive disulphide bridge. Our findings indicate that the activation of the thiol-switch in the circulating maternal angiotensinogen occurs at the placental level in response to oxidative stress, exacerbated by placental insufficiency. We propose here that a contributory factor is the inherent redox stress accompanying the placental exchange of oxygenation between the haemoglobin of the mother (oxy-HbA) and the deoxygenated haemoglobin of the foetus (deoxy-HbF).

## 1. Introduction

We outline here the new understandings at a structural molecular level of the central role of angiotensinogen in the onset of episodic hypertension, and notably of pre-eclamptic hypertension. In doing so, we also address, with a new proposal, the question as to the nature of the oxidative stress underlying the episodic onset, and of its placental focus in pregnancy.

## 2. Angiotensinogen and Renin

Angiotensinogen is a member of the serpin family of protease inhibitors that has lost its inhibitory activity and evolved to function as the carrier in the blood of the vasopressor peptides that control blood pressure [1,2,3]. The precursor of these peptides is angiotensin-1, a decapeptide at the amino-terminus of the extensive tail of angiotensinogen, which is cleaved from the tail by the highly specific protease renin. As indicated in Figure 1, the released angiotensin-1 is then further cleaved to produce the octapeptide angiotensin-2 and the smaller subpeptides that influence salt retention and control blood pressure [4,5]. Earlier studies on hypertension have focused on this subsequent cleavage of angiotensin-1 by the angiotensin-converting enzyme ACE, with the role of angiotensinogen being initially relegated to that of a passive substrate. Further studies, however, have indicated a direct involvement of angiotensinogen itself in the modulation of angiotensin release, with the rate-limiting step in the whole process being the cleavage of angiotensinogen by renin [6,7]. Direct evidence of the key role of this initiating cleavage in the control of blood pressure was found [8] in a family with a mutation at the renin cleavage site L10F in angiotensinogen. The mutation resulted in an increased efficiency of release of the precursor angiotensin-1 peptide and was associated with the onset of hypertension and, critically, the hypertensive crisis of pregnancy, pre-eclampsia.

## 3. Structural Mechanism of Cleavage and Angiotensin Release

Clarification of the mechanisms involved in the release of angiotensin came with solving a series of crystallographic structures of angiotensinogen, including its derivatives and complexes with renin. The initial structures [9] immediately confirmed that angiotensinogen had retained the typical serpin framework structure (Figure 1), with a striking difference being the presence in angiotensinogen of an extended 63-residue amino-terminal tail. The tail, which contained at its terminus the angiotensin sequence, was anchored to the body of the molecule, with the renin cleavage site Leu10-Val11 held in an inaccessibly buried position.

Further high-resolution structures [10,11] revealed the profound changes in shape that take place as renin approaches angiotensinogen to allow the precise docking of the two molecules (Figure 2; video). The CD loop of angiotensinogen, which otherwise blocks the binding of renin, undergoes a 10–20 Å displacement, with a linked extension of the angiotensin-containing tail and the entry of its scissile bond into the active site of renin. The terminal segment of the tail containing the angiotensin peptide is further stabilised for cleavage and subsequent release by a tail-in-mouth allosteric mechanism, with its reversible binding to what, in other serpins, is a well-defined hormone-binding site [10].

The video depiction shows [12] the choreographed shifts involved in the docking of the two molecules, with a displacement of the CD loop containing Cys138 and a linked extension of the amino-terminal tail containing Cys18. An accompanying unfolding of helix H opens an interface with renin that optimally positions the scissile site for the cleavage and release of the angiotensin.

## 4. Disulphide Modulation

The series of structures also highlights the key role of the conserved disulphide bridge, Cys18-138, in the coordination of the steric changes involved in the interaction with renin. Independent biochemical evidence of the critical function of this disulphide bridge between Cys18 in the amino-terminal tail of angiotensinogen and Cys138 in the CD loop of the body of the molecule came earlier [13,14], with the observation of the constraint on the movement of the angiotensin tail in oxidised (S-S bridged) angiotensinogen. The finding that this bridging profoundly influences the kinetics of interaction with renin [13] is in keeping with the subsequent recognition of the disulphide modulation of angiotensin release.

The 18–138 disulphide in angiotensinogen was shown [9] to be labile, with a redox poise allowing a ready equilibration between its oxidised bridged form and its reduced unbridged form, as demonstrated by the consistent presence in the plasma of the two forms at a reduced-to-oxidised ratio near 40:60, independent of gender or age (Figure 3). The modulatory function of the disulphide bridge is apparent in the fourfold increase in the catalytic release of angiotensin that results from the switch from the fully reduced to the fully oxidised forms [9]. Hence, the maintenance of this remarkably constant 40:60 redox ratio in plasma angiotensinogen is a likely requirement for normotensive health.

The deduced modulatory role of the 18–138 disulphide is in keeping with repeated reports that episodic hypertension can be triggered by oxidative stress [15,16,17,18], which favours a transition of angiotensinogen to its more active bridged form. Inferential support for this modulatory function of the bridge comes from the demonstration [9] of the ready reduction of the 18–138 disulphide by S-nitrosylation, a sensitively selective process dependent on structural interactions and indicative of specific biological functions [19].

## 5. Pre-Eclampsia

The deduction that the onset of episodic hypertension could arise from the oxidative conversion of angiotensinogen to its more active bridged form was difficult to experimentally confirm, as the conversion is likely to take place diffusely at the tissue level. There is, however, a notable exception to this in pre-eclamptic hypertensive crises, which often complicate the later stages of pregnancy and are a leading cause of maternal and neonatal morbidity and mortality [20]. Here, the oxidative exposure of the circulating angiotensinogen of the mother occurs within the focused intervillous spaces and milieu of the placenta. Thus, any consequent modifications that occur to the disulphide bridge in pre-eclampsia should be detectable on plasma sampling. As shown in Figure 3, this proved to be so [9]. A blinded analysis of maternal plasma samples from 12 patients with pre-eclampsia with carefully matched normal pregnancy controls showed self-evident increases in the proportion of oxidised angiotensinogen in the pre-eclamptic plasmas. These findings have since been confirmed by other independent assays in patients with pre-eclampsia, showing similar increases in the proportion of oxidised angiotensinogen [21,22], together with a decrease in plasma antioxidants [23].

Convincing evidence that the fourfold increase in the catalytic efficiency of cleavage of oxidised angiotensinogen [9] is a sufficient cause for pre-eclamptic hypertension comes from Inoue and colleagues’ study of their index case of a 17-year-old with pre-eclampsia [8]. In this case, a mutation at the renin cleavage site in angiotensinogen L10F resulted in an increased catalytic efficiency of cleavage by renin, comparable to the change observed here, with the transition of angiotensinogen from the reduced to oxidised form. Eclampsia itself is an umbrella term, covering what is probably a range of aetiologies. This index case, however, completes a chain of evidence linking the onset of pre-eclamptic hypertension to the observed change in the cleavage kinetics of angiotensinogen following its oxidative exposure in the placenta.

## 6. Circulatory Milieu: Haemoglobin a Redox Agent?

The stable redox milieu of the circulation maintains the modulatory disulphide in angiotensinogen, with an optimal equilibrated ratio between its reduced open form and its oxidised bridged form of 40:60 (Figure 3a). This equilibrium breaks down, however, in pre-eclampsia, which is often referred to as a disease of the placenta. Thus, any proposal as to the underlying causation of pre-eclampsia must explain the associated increased proportion of the more active oxidised form of angiotensinogen (Figure 3b) and, critically, its origin from focal redox stress in the placenta of late pregnancy.

A clue to a likely contributory cause to this focal stress comes from much earlier studies on haemoglobin breakdown and red cell oxidation [24,25]. Notably, the red cell studies revealed—as we propose is also relevant in pre-eclamptic hypertension—how redox stress may inherently accompany the deoxygenation of haemoglobin [26,27]. An early indication that oxygen was bound to the iron of the haem group of haemoglobin, not as a molecule of oxygen but as the superoxide anion, came in the aftermath of solving the crystallographic structure of myoglobin [28,29], from the oblique orientation of oxygen to the iron of the haem. Direct evidence that the bound oxygen could be aberrantly released from haemoglobin as the superoxide radical followed from studies on the autoxidation of haemoglobin [30] and the consequences of globin instability [27]. The overall findings opened the concept of the function of the haem moiety as an electron donor, with the binding and release of oxygen involving the reversible polarisation of an electron between the iron of the haem group and the divalent oxygen [31].

This concept of the haem as an electron source was widened in follow-up studies of a uniquely conserved cysteine, present in the haemoglobin beta-chain in all species, β93Cys. This thiol was well known to readily bond to a range of agents, but this only occurred when the haemoglobin was oxygenated and not with deoxyhaemoglobin. The dogma then was that the cysteine was sterically inaccessible in deoxyhaemoglobin. This assumption was overturned in the follow-up of an observation in 1976 that copper could bind to the β93Cys and induce the oxidation of the haem iron [32]. A viewing of the newly refined structure of haemoglobin [33] by M. F. Perutz showed that, contrary to earlier expectations, the β93Cys was fully accessible in deoxyhaemoglobin. Examination of the structure also revealed a hydrophobic pathway that would allow charge transfer from the exposed thiol to the iron of the haem [32]—a charge transfer that, on the oxygenation of haemoglobin, would activate the thiol of the β93Cys.

The vital physiological function of this linked activation of the beta 93 thiol became evident in the proposal by Stamler and colleagues [19,34,35,36,37] of an alternative circulatory role of haemoglobin as a principal carrier and distributor of the ubiquitous signalling radical nitric oxide (NO*). Nitric oxide is bound in stable form to the β93Cys of haemoglobin as SNO, with the binding readily taking place with the activation of the β93Cys in oxyhaemoglobin and with an equivalent release of SNO on deoxygenation. Physiologically, haemoglobin circulates throughout the body with a tissue oxygenation of some 70% saturation, such that an exchange of oxygen between oxy- and deoxy-haemoglobin molecules is a constant process, with an accompanying equivalent uptake and release of SNO. Any balance of the SNO released on the deoxygenation of haemoglobin is taken up, in an appropriate response to hypoxia, by the endothelium of the microcirculation to give vasodilation and a compensating increase in blood flow.

The circulating S-nitrosylated haemoglobin (SNO-Hb) also has another critical biological function. This is a broader modulatory role in its deoxy form as an S-nitrosotransferase [19], interacting selectively with the tissue disulphides that control a range of key functions and, specifically, with the 18–138 disulphide in angiotensinogen [9] (Figure 3d and Figure 4).

## 7. Placental Oxidative Stress

A notable exception to the balanced redox milieu of the blood occurs at one of the body’s major sites of oxygen exchange, the placenta during pregnancy. Here, the release of SNO that accompanies the release of oxygen from the maternal oxyhaemoglobin is not balanced by the transfer of the SNO to the deoxyhaemoglobin of the foetal circulation. The gamma globin of HbF lacks the equivalent of the 93Cys of the beta-globin of HbA, and hence, it cannot bind or carry the released SNO. Although only one or two out of every thousand beta-globin molecules is thiolated [19], the concentration of haemoglobin in the body is so high that the placenta, with this unbalanced release of SNO, potentially becomes a nitrosylation ‘pool’. This exposure of maternal plasma proteins to nitrosylation is exacerbated by the circuitous flow of maternal blood through the placenta. This flow does not take place briskly, as in the arterial capillaries of the tissues, but rather in the more measured sinusoidal flow of the intervillous spaces of the placenta. An additional threat to direct exposure to SNO comes from the action of the freshly deoxygenated maternal haemoglobin as an S-nitrosotransferase, specifically interacting with the 18–138 disulphide bridge in angiotensinogen (Figure 3e). The overall outcome and clinical correlations of this placental exposure, with respect to angiotensinogen, are seen in Figure 3a–d. The consumption and deterioration of the maternal angiotensinogen in eclampsia is evident, notably as enhanced ‘oxidation’.

Although we focus here on angiotensinogen, the exposure of the maternal plasma also affects its other components, and relevantly so, blocking the principal reductant in blood, glutathione [37,38,39]. Added to the release of SNO, there is also an awareness of the broader binding capabilities of the β93Cys and the potential carriage and placental release of other as yet unidentified radicals. Taken together, these factors contribute to placental redox stress that activates the modulating disulphide of angiotensinogen and hence underlies the development of the hypertension of pre-eclampsia.

## 8. Conclusions

We outline here, in molecular detail, the mechanisms leading to the pre-eclamptic hypertensive crises of pregnancy. Although pre-eclampsia is an umbrella diagnosis, as shown in Figure 3c,d, it is agreed that it principally arises from still-undefined oxidative challenges [15,16,17,18]. Direct support for this is the predominant finding in pre-eclampsia (Figure 3b) of oxidative modifications of the disulphide bridge that regulates the release of the precursor angiotensin peptide.

That pre-eclampsia uniquely develops in humans compared to other primates has been attributed to the deep trophoblastic invasion of the placenta necessary to meet the oxygen demands for the growing brain in the human foetus [40]. The expansion of the trophoblast carries with it a risk of vascular stasis and hence localised redox stress. The main safeguard against the aggregation of this stress is the buffering and sweeping away provided by efficient circulatory flow. The vulnerability posed by reduced blood flow in the trophoblastic circulation of the human placenta fits with recurrent reports of the association of the onset of pre-eclampsia with placental insufficiency and reduced perfusion [41].

We propose here that an underlying contributor to the placental redox stress arises from the convergence of the adult and foetal haemoglobins. If such redox stress is an inherent risk from the mixing of the maternal and foetal circulations in late pregnancy, then would we expect to see other inbuilt mechanisms to alleviate or counter this? To a degree, this is so, in that the foetus, as it matures and its oxygen demands increase, also begins to switch its synthesis of haemoglobin from the susceptible foetal HbF to adult HbA, so that by the third trimester, HbA forms some 20% of the haemoglobin of the maturing foetus. There are, however, considerable individual variations in the timing and magnitude of this switch, with further validation coming from clinical reports of an association of the onset of pre-eclamptic hypertension with a persistent predominance in HbF expression [42,43,44].

Although the findings presented here describe in detail the molecular pathology of pre-eclamptic hypertension, they leave open the question of the initiating causes. A vast amount of research with accompanying literature has addressed this question, and numerous subtypes of the eclampsia syndrome have been identified [45,46]. A consensus emerging from this is that pre-eclampsia is a placental disease, or rather, a consequence of placental dysfunction. This is consistent with the manifestations we detail here, with episodic hypertension arising from uncompensated shortcomings of the placenta in dealing with oxidative stress.

Figure 2, Figure 3 and Figure 4 were adapted with acknowledgement from Ref. [9]: Nature 468, 2010, 108–111 and its Supplementary Materials.

## Figures and Tables

**Figure 1 cimb-47-00375-f001:**
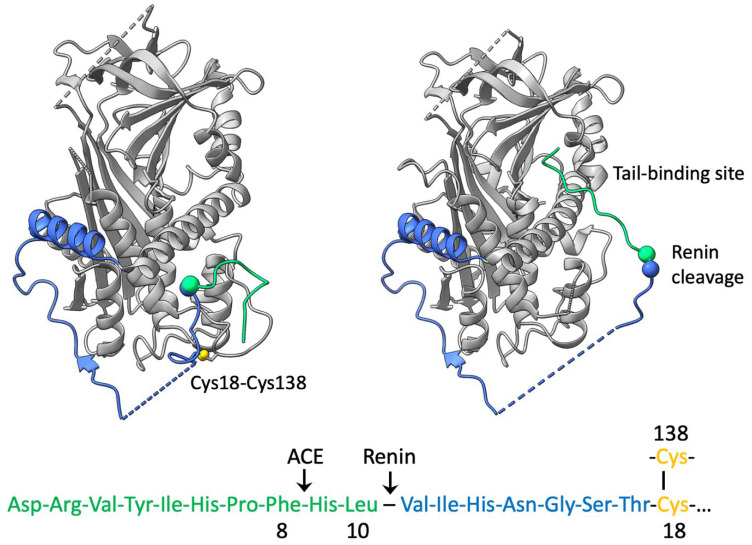
Structures of angiotensinogen [9] confirm its typical serpin fold (grey), with, on the (**left**), the extended tail in blue and the terminal angiotensin sequence and its buried cleavage point (ball) in green. The 18–138 disulphide linking the tail to the CD-loop of the body of the molecule is in yellow. On the approach of renin (**right**), the CD loop moves, and the tail is extended, exposing the cleavage site. The terminal portion of the tail binds [10] to a site, as shown, causing a localised unfolding and revealing residues critical to the binding of renin and preferential release of the cleaved angiotensin.

**Figure 2 cimb-47-00375-f002:**
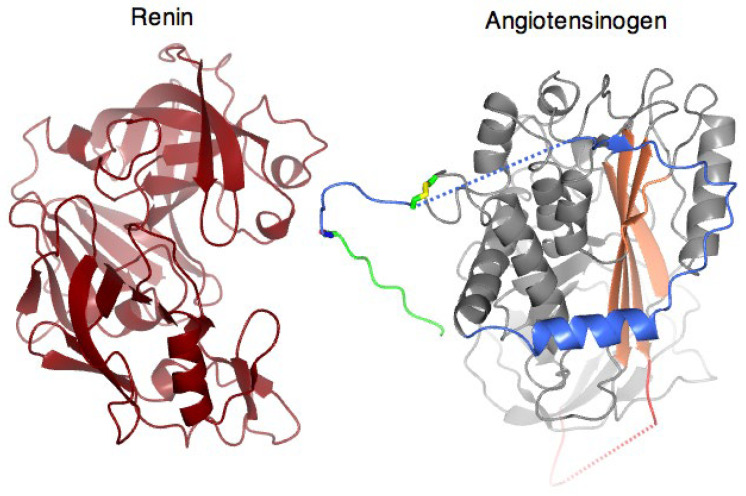
The video demonstrates the conformational changes that occur when human angiotensinogen binds to human renin. Renin is shown on the (**left**) in brown, and angiotensinogen is shown on the (**right**), mostly in grey, with the N-terminal angiotensin I peptide highlighted in green, the remainder of the N-terminal extension in blue, the reactive centre loop in red, the Cys18-138 in yellow, and the main A-sheet in coral [12]. https://static-content.springer.com/esm/art%3A10.1038%2Fnature09505/MediaObjects/41586_2010_BFnature09505_MOESM78_ESM.mov (accessed on 14 April 2025).

**Figure 3 cimb-47-00375-f003:**
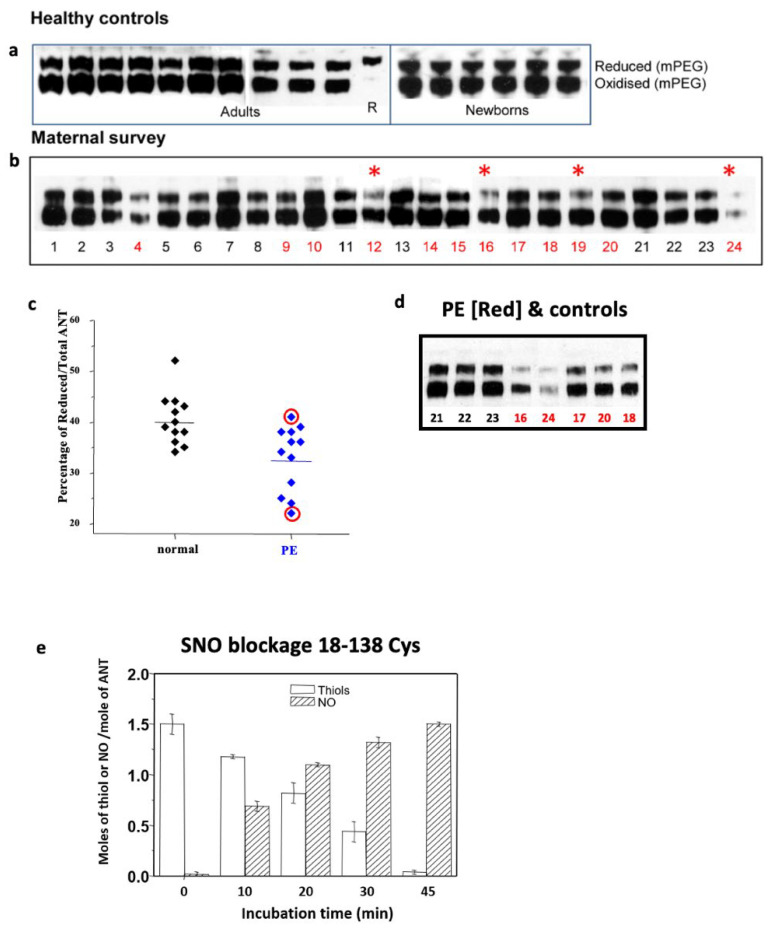
(**a**) Angiotensinogen oxidation. Western blot SDS-PAGE of plasma exposed to N-ethylmaleimide-polyethylene-glycol adduct (mPEG). Samples from healthy controls show remarkably constant 40:60 proportions of the reduced to oxidised forms, as do cord plasma from healthy newborns. (**b**) Maternal plasma samples from patients with pre-eclampsia and matched normotensive controls were sequentially referred to with blinded coding. The recognisably lower proportion of reduced angiotensinogen in pre-eclamptic plasmas was confirmed upon breaking the coding (normotensive black numbering; patients with pre-eclampsia red. Red asterisks mark the four patient samples below the mean in (**c**). (**c**) Reduced/total angiotensinogen percentages. Distribution of percentages of the reduced form of angiotensinogen in patients with pre-eclampsia (blue) and normotensive controls (black). (**d**) Clinical correlations. Pre-eclampsia is an umbrella title covering more than one pathology. The uppermost sample ringed in (**c**) is from a patient with pre-eclampsia, but the gel shows a typically normal pattern with no decrease in reduced angiotensinogen. Although the sample comes from a pregnant woman with the defining criteria for pre-eclampsia—hypertension and proteinuria—she differed from other cases in fitting with a pregnancy-induced presentation of latent essential hypertension rather than the typical oxidatively-induced hypertension of pre-eclampsia. The lowermost sample, ringed in c, and sample 24 in the gel in d come from the most severely affected case, with onset pre-eclampsia requiring delivery at 25 weeks. The reanalysis of this sample demonstrates not only the relative decrease in the reduced form but also clearly decreased levels of angiotensinogen, which is also apparent, though less evident, in the other pre-eclamptic samples (red numerals). The concentrations of angiotensinogen here are in keeping with an increased consumption that would predictably result from the activation of angiotensinogen in pre-eclampsia [9]. Note. The term ‘oxidative changes’ is generic; as used here, it refers to the lower-molecular-weight component seen on electrophoresis, subsequent to the mPEG tagging of free thiols, as shown in Figure 3. In good health, this component represents the bridged form of angiotensinogen, but under redox stress, as in eclampsia, it also includes nitrosylated and other ligand-blocked forms. (**e**) S-nitrosothiols (NO). The ready blocking and conversion of the two SH (thiol) groups of the modulatory 18–138 disulphide of angiotensinogen, on incubation at 37 °C with an S-nitrosothiol donor [9].

**Figure 4 cimb-47-00375-f004:**
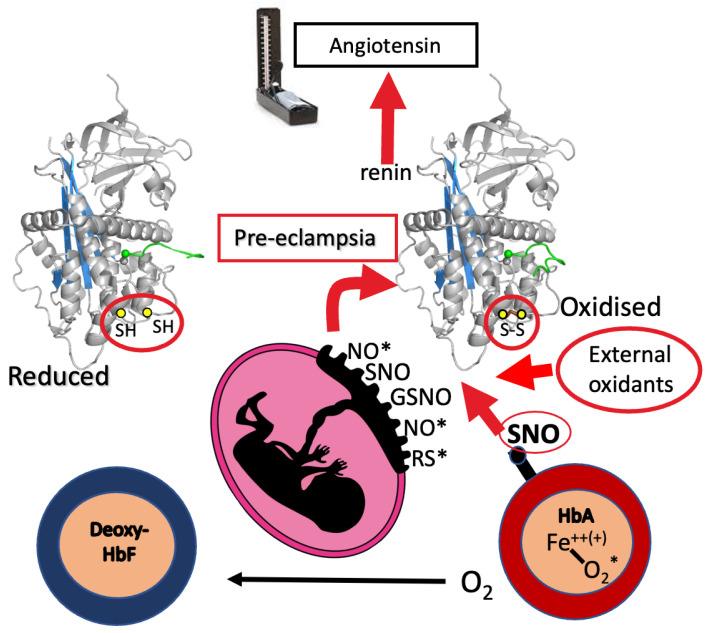
Schematic. Angiotensinogen oxidation and hypertension. Angiotensinogen, the plasma protein source of the peptides that control vasoconstriction, exists in equilibrium between its oxidised form (on the right) and its reduced form (on the left). In pre-eclampsia, the balance shifts to the more active oxidised form, with a consequent increase in angiotensin release, leading to the episodic hypertension of pre-eclampsia. We propose that a contributory cause of the placental redox stress arises from the release of oxidising radicals that inherently takes place on the convergence of the maternal HbA circulation with the HbF of the foetal circulation. The HbA of the adult has an active thiol, β93Cys, that functions as a carrier of nitric oxide (NO*), bound to the thiol as SNO when the haemoglobin is oxygenated but then released on deoxygenation. The foetal haemoglobin HbF lacks this carrier cysteine and hence cannot take up the SNO released from the maternal circulation when the massive exchange of oxygen between the two circulations occurs in the placenta. Oxidising radicals released at the placenta are indicated by the symbols NO*, SNO, GSNO (S-nitrosoglutathione) and RS* (other released radicals).

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
