# Peer review of "The Molecular Pathology of Pre-Eclamptic Hypertension"

_cimb, 2025, doi:10.3390/cimb47050375_

Round 1
Reviewer 1 Report
Comments and Suggestions for Authors
This is an interesting manuscript with quite adequate novelty. However, some points should be addressed.
- The Abstract is quite simple and not enough representative of the following text. More information should be added in this section.
- The basic aim of the study should be added at the end of the Introduction section.
- The sentence in lines 82-84 ''Maintenance of this ratio is a likely requirement for good health as the switch from the reduced to the oxidised-form results in a fourfold increase in the catalytic release of angiotensin.'' needs a bit more analysis.
- The resolution and the size of Figure 3 should be increased.
- The sentence in lines 127-130 ''As indeed proved to be so: evident changes in angiotensinogen were clearly seen (Fig 3) in maternal plasma samples from pre-eclamptic pregnancies with consistent increases in the oxidised form as compared to carefully matched normal-pregnancy controls[9]." is quite complex and needs rephrasing in order to be more easily understood by the readers.
- Again, the sentence in lines 134-137 ''That the fourfold increase of the catalytic efficiency of cleavage of oxidised angiotensinogen[9] is a sufficient cause of pre-eclamptic hypertension is confirmed by its comparability in magnitude to the change observed in the index case of the 17-year old with familial (L10F) pre-eclampsia[8]." is quite complex and needs rephrasing in order to be more easily understood by the readers.
- Again, the sentence in lines 194-199 ''Here the release of SNO that accompanies the release of oxygen from the maternal oxyhaemoglobin, will not be balanced by transfer of the SNO to the deoxyhaemoglobin of the fetal circulation, as the gamma globin of fetal haemoglobin lacks the carrier 93Cys, singularly present in the beta subunits of adult haemoglobin." is quite complex and needs rephrasing in order to be more easily understood by the readers.
- English language editing is highly recommended.
- There are several typos and syntax/grammar errors throughout the manuscript that should be revised.
- English language editing is highly recommended.
- There are several typos and syntax/grammar errors throughout the manuscript that should be revised.
Author Response
We thank the reviewers for their helpful comments.
Comments1: The Abstract is quite simple and not enough representative of the following text. More information should be added in this section.
Response1: Rewritten within word limitations
Comments2: The basic aim of the study should be added at the end of the Introduction section.
Response2: Done
Comments3: The sentence in lines 82-84 ''Maintenance of this ratio is a likely requirement for good health as the switch from the reduced to the oxidised-form results in a fourfold increase in the catalytic release of angiotensin.'' needs a bit more analysis.
Response3: Rephrased
Comments4:The resolution and the size of Figure 3 should be increased.
Response4: Done
Comments5: The sentence in lines 127-130 ''As indeed proved to be so: evident changes in angiotensinogen were clearly seen (Fig 3) in maternal plasma samples from pre-eclamptic pregnancies with consistent increases in the oxidised form as compared to carefully matched normal-pregnancy controls[9]." is quite complex and needs rephrasing in order to be more easily understood by the readers.
Again, the sentence in lines 134-137 ''That the fourfold increase of the catalytic efficiency of cleavage of oxidised angiotensinogen[9] is a sufficient cause of pre-eclamptic hypertension is confirmed by its comparability in magnitude to the change observed in the index case of the 17-year old with familial (L10F) pre-eclampsia[8]." is quite complex and needs rephrasing in order to be more easily understood by the readers.
Again, the sentence in lines 194-199 ''Here the release of SNO that accompanies the release of oxygen from the maternal oxyhaemoglobin, will not be balanced by transfer of the SNO to the deoxyhaemoglobin of the fetal circulation, as the gamma globin of fetal haemoglobin lacks the carrier 93Cys, singularly present in the beta subunits of adult haemoglobin." is quite complex and needs rephrasing in order to be more easily understood by the readers.
English language editing is highly recommended.
Response5: All rewritten in response to these comments.
Comments6: There are several typos and syntax/grammar errors throughout the manuscript that should be revised.
Response6: Proofread and corrected.
Reviewer 2 Report
Comments and Suggestions for Authors
This is an interesting review. My main criticism concerns the quality of the manuscript's written content, as several sections are difficult to follow. I strongly suggest a careful revision of the entire manuscript to enhance readability and clarity. For instance, the abstract should be rewritten to be clearer and precise. Additionally, the Clinical Summary or message for clinicians contains expressions such as "smoking gun" and "magic bullet" that seem inappropriate for this type of review. This section, in particular, does not offer new, precise information for clinicians.
Furthermore, the authors must take into account that several hypotheses exist regarding the causes of preeclampsia. However, the precise mechanisms initiating and sustaining this syndrome remain unclear. Current understanding suggests that preeclampsia is triggered either by specific trophoblast defects (of placental origin, also known as early-onset preeclampsia) or by maternal metabolic defects (of maternal origin, also known as late-onset preeclampsia). As previously mentioned, the placenta releases various factors into the maternal circulation. The interplay of these factors, along with the maternal response and susceptibility, determines the onset, severity, fetal involvement, and progression of preeclampsia. Numerous studies have characterized and identified various features of this clinical syndrome, leading to the definition of specific subtypes. This is important because the authors of this review focus solely on the placental origin of the underlying oxidation in preeclampsia.
Comments on the Quality of English LanguageAs mentioned above, I would suggest revising the quality and the content of the review to enhance its readability and clarity.
Author Response
Comments1: This is an interesting review. My main criticism concerns the quality of the manuscript's written content, as several sections are difficult to follow. I strongly suggest a careful revision of the entire manuscript to enhance readability and clarity. For instance, the abstract should be rewritten to be clearer and precise. Additionally, the Clinical Summary or message for clinicians contains expressions such as "smoking gun" and "magic bullet" that seem inappropriate for this type of review. This section, in particular, does not offer new, precise information for clinicians.
Response1: Revision carried out. The Clinical Summary has been removed.
Comments2: Furthermore, the authors must take into account that several hypotheses exist regarding the causes of preeclampsia. However, the precise mechanisms initiating and sustaining this syndrome remain unclear. Current understanding suggests that preeclampsia is triggered either by specific trophoblast defects (of placental origin, also known as early-onset preeclampsia) or by maternal metabolic defects (of maternal origin, also known as late-onset preeclampsia). As previously mentioned, the placenta releases various factors into the maternal circulation. The interplay of these factors, along with the maternal response and susceptibility, determines the onset, severity, fetal involvement, and progression of preeclampsia. Numerous studies have characterized and identified various features of this clinical syndrome, leading to the definition of specific subtypes. This is important because the authors of this review focus solely on the placental origin of the underlying oxidation in preeclampsia.
Response2: Point taken. We discuss other aspects in the final paragraphs of the paper.
Round 2
Reviewer 1 Report
Comments and Suggestions for Authors
The manuscript has sufficiently been improved after the revisions by the authors.
Reviewer 2 Report
Comments and Suggestions for Authors
I do not have further comments. This is a better version of the manuscript.